# Hyperspectral-Based Estimation of Leaf Nitrogen Content in Corn Using Optimal Selection of Multiple Spectral Variables

**DOI:** 10.3390/s19132898

**Published:** 2019-06-30

**Authors:** Lingling Fan, Jinling Zhao, Xingang Xu, Dong Liang, Guijun Yang, Haikuan Feng, Hao Yang, Yulong Wang, Guo Chen, Pengfei Wei

**Affiliations:** 1National Engineering Research Center for Agro-Ecological Big Data Analysis & Application, Anhui University, Hefei 230601, China; 2Beijing Engineering Research Center of Agricultural Internet of Things, Beijing 100097, China

**Keywords:** hyperspectral, leaf nitrogen content (LNC), successive projections algorithm (SPA), partial least squares (PLS) model, random forest (RF) model

## Abstract

Accurate and dynamic monitoring of crop nitrogen status is the basis of scientific decisions regarding fertilization. In this study, we compared and analyzed three types of spectral variables: Sensitive spectral bands, the position of spectral features, and typical hyperspectral vegetation indices. First, the Savitzky-Golay technique was used to smooth the original spectrum, following which three types of spectral parameters describing crop spectral characteristics were extracted. Next, the successive projections algorithm (SPA) was adopted to screen out the sensitive variable set from each type of parameters. Finally, partial least squares (PLS) regression and random forest (RF) algorithms were used to comprehensively compare and analyze the performance of different types of spectral variables for estimating corn leaf nitrogen content (LNC). The results show that the integrated variable set composed of the optimal ones screened by SPA from three types of variables had the best performance for LNC estimation by the validation data set, with the values of R^2^, root means square error (RMSE), and normalized root mean square error (NRMSE) of 0.77, 0.31, and 17.1%, and 0.55, 0.43, and 23.9% from PLS and RF, respectively. It indicates that the PLS model with optimally multitype spectral variables can provide better fits and be a more effective tool for evaluating corn LNC.

## 1. Introduction

Hyperspectral data with high spectral resolution could reveal small changes in the biochemical components of plant leaves, and its acquisition was rapid and non-destructive. In fact, rapid, non-destructive monitoring of plant leaf biochemical components by hyperspectral means has now became an important part of the evaluation of vegetative growth status. However, although hyperspectral data with hundreds or even thousands of bands could provide more detailed and richer spectral information than multispectral data, it suffered from significant data redundancy, high correlation between adjacent bands, etc. [1]. Therefore, more research was required to determine how best to extract from hyperspectral data the characteristic spectral variables to effectively monitor the biochemical components of crop targets.

At present, spectral variables based on extracted hyperspectral data might be divided into three categories generally:

(1) Characteristic reflected bands. Hyperspectral data offer more accurate bands, which can better reflect the characteristics of vegetation. Bai et al. [2] applied the successive projections to extract eight sensitive bands correlated with total nitrogen content of winter wheat leaves (1985, 2474, 1751, 1916, 2507, 1955, 2465, and 344 nm). These bands had an extremely negative correlation, great significance and established a highly accurate and stable successive projections algorithm- partial least squares (SPA-PLS) model to estimate leaf content in the wheat jointing stage. He et al. [3] used the variable importance for projection (VIP) and the grey relational analysis (GRA) to extract five optimal first derivative spectra in the range of 350–2500 nm in winter wheat. On the whole, those bands had a pretty strong relationship and established better results with leaf nitrogen content (LNC) in validation.

(2) Position of reflected features. Reflectance and absorption features that characterize hyperspectral data are also related to specific physical and chemical crop characteristics [4]. Wei et al. [5] extracted the red edge position by using six different methods, and analyzed the relationship between the red edge position extracted from canopy spectra and associated LNC of the vegetation above ground. Cho et al. [6] extracted the red edge position (REP) from rye canopy, and corn leaf and mixed grass/herb leaf stack hyperspectral data via a new technique and REPs extracted using this new technique (linear extrapolation method) showed high correlations with a wide range of foliar nitrogen concentrations (NC).

(3) Vegetation index characteristics. Vegetation indices are combinations of linear and nonlinear characteristics in the visible-near-infrared band and quantitatively reflect vegetation growth under certain conditions. Chen et al. [7] compared the double-peak canopy nitrogen index (DCNI), with some existing vegetation indices such as modified chlorophyll absorption ratio index (MCARI), canopy chlorophyll index (CCI), Medium Resolution Imaging Spectrometer (MERIS) terrestrial chlorophyll index (MTCI), etc. and determined that the DCNI of wheat and corn provided the best spectral index for evaluating the efficiency of crop nitrogen treatment. Finally, through the correlation analysis among the normalized differential red edge index (NDRE), water sensitivity index (WI), and crop leaf area index (LAI). Shu et al. [8] constructed a new red-edge resistance water vegetable index (RRWVI) to improve the accuracy of hyperspectral inversion of the crop leaf area index (LAI). Tan et al. [9] comprehensively analyzed the correlation and predictability of ratio vegetation index (RVI), normalized difference vegetation index (NDVI), difference vegetation index (DVI), etc. and summer corn LAIs. The results suggested that the correlation between those commonly used spectral vegetation indices and LAI reached the significant level of 0.05.

On the one hand, in this paper, the characteristics of corn canopy spectra were systematically summarized and studied from three perspectives. Those studies demonstrate that hyperspectral variable features have been utilized in research. Although these features could be used to monitor and evaluate crop growth parameters, most studies only use a single type of spectral variable. The various types of spectral variables provide useful information from different viewpoints about crop growth parameters. Therefore, if making use of different spectral variables to improve the accuracy offered a possible way with which crop-target parameters were monitored, little information was currently available on how to comprehensively exploit multiple hyperspectral variables to better exploit the rich information of hyperspectral spectrum and thereby we might improve the accuracy of crop nutrient status. On the other hand, hyperspectral data were characterized by multiple collinearity. Multiple linear regression analysis (MLRA) model, PROperties SPECTra (Prospect) model, Decision Support System for Agrotechnology Transfer (DSSAT) model, and so on were commonly used methods for LNC inversion. The PLS regression method was an extension of the multiple linear-regression model, which was widely used because it could reduce the problem of collinearity between data variables [10,11,12]. Furthermore, although the random forest (RF) model, which was used mostly in biology and had high predictive and learning ability, resolved the problem of singular values between response variables and explanatory variables [13], few reports had used it to monitor the nitrogen content in corn. Thus, the present study applied these three categories of spectral variables based on extracted hyperspectral information and combines them with sensitive variables selected by using the successive projections algorithm (SPA) to estimate the LNC of corn. Furthermore, we compared the PLS and RF modeling methods for monitoring corn LNC to obtain new ideas and methods for evaluating the nitrogen nutrition spectrum of crops.

## 2. Materials and Methods

### 2.1. Study Area and Preprocessing

#### 2.1.1. Study Area

The experiment was done at the National Precision Agriculture Research and Demonstration Base in 2012. The base is located northeast of Xiaotangshan Town, Changping District, Beijing (40°00′–40°21′ N, 116°34′–117°00′ E, 36 m). The climate was a temperate continental monsoon climate. The soil in the experimental area was a silt-clay loam, which the PH value reached 8.0. The average soil nutrients of the site were as follows: Organic matter 1.14%; alkaline nitrogen 49.9 mg·kg^−1^; available phosphorus 17.0 mg·kg^−1^; and available potassium 145 mg·kg^−1^. Three nitrogen levels were used: No nitrogen content of 0 kg N**·**ha^−1^, normal nitrogen content of 337 kg N**·**ha^−1^, and excess nitrogen content of 765 kg N**·**ha^−1^. The experiments were done in replicates of three. We used 18 study plots. 1#–9# plots were planted ‘Nongda 108′ (Mid-drape type) and 10#–18# plots were planted ‘Jinghua 8′ (compact type). The total study area was 924 m^2^ and each area was 7 × 7 m^2^. The blank line was 42 m^2^ in the middle. Figure 1 shows the study plots. Corn samples were extracted at four growth stages of V6, V14, R1, and R2 in 18 plots, respectively. Sowing was done on 21 June 2012 and harvesting on 15 October 2012 of corn. All plots followed the local standard practices (weed control, pest management, and fertilizer application).

#### 2.1.2. Spectrum Acquisition 

The experiment was done during the different growth stages of corn: July 30 (V6—6 leaf), August 15 (V14—14 leaf), August 26 (R1—silking), and September 14 (R2—blister stage). Corn canopy hyperspectral data were collected from each experimental plot.

The spectral reflectance was acquired by using an ASD FieldSpec FR2500 Spectrometer (Analytical Spectral Device, Boulder, CO, USA) with a spectral range of 350–2500 nm. The resolution was 1.4 nm from 350 to 1000 nm, and 1 nm from 1000 to 2500 nm. Generally, measurements were done at 10:00 a.m. and 14:00 p.m. Beijing time during clear, windless, cloudless conditions. The probe was oriented vertically downward when viewed. The height was 1.3 m from the ground and the field of view angel was 25°. Each measurement was corrected before and after by using the reference plate.

#### 2.1.3. Plant Sample and LNC Acquirement

After making spectral measurements of each experimental plot, the stems and leaves were separated, and the leaves were placed in a paper bag. The leaves were then placed in an oven at 105 °C for 30 min, and then baked at 80 °C for 48 h or more until weighed. The dried-leaf samples were weighed, and then the leaves were pulverized and their nitrogen content was determined by using a Kjeldahl analyzer (Buchi B-339, FOSS, Sweden). The total statistics were 72 of four growth stage in 18 plots; 48 for calibration, and 24 for validation. Table 1 showed the total statistics of green LNC measured. The LNC range for the calibration dataset in 2012 was from 0.92 to 2.83, with an average of 1.91 and a standard deviation of 0.59. Similarly, the statistical parameters for the test dataset in 2012 was 0.82–2.68, 1.81, and 0.65, respectively.

### 2.2. Principles and Methods

#### 2.2.1. Preprocessing of Hyperspectral Data

To eliminate part of the noise in the spectrum, we applied a Savitzky-Golay (SG) convolution smoothing method [14,15]. Based on the results of preliminary experiments, maximum denoising was achieved with a moving window width of 17 and a polynomial frequency of two. We calculated various spectral variables, such as first derivative (FD), position features, and vegetation indices, based on the spectral reflectance after SG denoising. The FD formula was:(1)$${\mathrm{FD}}_{\lambda \left(i\right)}=\frac{R_{\lambda \left(j+1\right)}-R_{\lambda \left(j\right)}}{\lambda \left(j+1\right)-\lambda \left(j\right)},$$
where FD is the first derivative of reflectance at wavelength midpoint *i* between wavebands *j* and *j* + 1, $R_{\lambda \left(j\right)}$ is the reflectance at waveband *j*, $R_{\lambda \left(j+1\right)}$ is the reflectance at waveband *j* + 1, and λ(j+1)−λ(j) is the difference in wavelength between wavebands *j* and *j* + 1.

#### 2.2.2. Spectral Position Features

Figure 2 showed the characteristic absorption and reflections of the summer corn canopy for three nitrogen treatments. The figure showed the three absorptions (560–760, 920–1080, and 1120–1280 nm) and six reflections (500–670, 780–970, 980–1200, 1200–1350, 1480–1720, and 2000–2300 nm) that were used to study the characteristic absorption and reflection positions [16]. In the present study, we explored only three parameters: Depth, area, and normalized depth [17,18].

The absorption depth ($A\_{\mathrm{Depth}}_{i}$) was calculated as follows:(2)$$A\_{\mathrm{Depth}}_{i}=1-{R^{\prime }}_{i}\left(\lambda_{\min}\right)=1-R_{i}\left(\lambda_{\min}\right)/R_{ci}\left(\lambda_{\min}\right),$$
where ${R^{\prime }}_{i}\left(\lambda_{\min}\right)$ is the continuum-removal reflectance and is defined as the ratio of $R_{i}\left(\lambda_{\min}\right)$ that is the reflectance at corresponding wavelength λ in the absorption region to the continuum line $R_{ci}\left(\lambda_{\min}\right)$ in the corresponding band. The index *i* identifies the number of absorption positions (*i* = 1, 2, 3).

The absorption area ($A\_{\mathrm{Area}}_{i}$) was calculated as follows:(3)$$A\_{\mathrm{Area}}_{i}=\int_{\lambda_{j}}^{\lambda_{k}}(R_{ci}\left(\lambda \right)-R_{i}\left(\lambda \right))d\lambda .$$
The absorption area ($A\_{\mathrm{Area}}_{i}$) is the integral of the difference between the reflectance of continuum line $R_{ci}\left(\lambda \right)$ and the reflectance $R_{i}\left(\lambda \right)$ at the corresponding wavelength λ in the absorption region. The wavelengths $\lambda_{j}$ and $\lambda_{k}$ are the initial and final wavelengths in each absorption region.

The normalized absorption depth ($A\_ND_{i}$) was:(4)$$A\_ND_{i}=A\_{\mathrm{Depth}}_{i}/A\_{\mathrm{Area}}_{i}.$$
$A\_ND_{i}$ is the ratio of the absorption depth to the integrated absorption wavelength.

Each reflection depth ($R\_{\mathrm{Depth}}_{i}$) was defined as: (5)$$R\_{\mathrm{Depth}}_{i}=1-{R^{\prime }}_{i}\left(\lambda_{\max}\right)=1-R_{ci}\left(\lambda_{\max}\right)/R_{i}\left(\lambda_{\max}\right).$$
The reflection depth is the difference between unity and the continuum-removed reflectance ${R^{\prime }}_{i}\left(\lambda_{\max}\right)$. The continuum-removed reflectance ${R^{\prime }}_{i}\left(\lambda_{\max}\right)$ is the ratio of the inner continuous line $R_{ci}\left(\lambda_{\max}\right)$ in the reflection position and the maximum reflectance value $R_{i}\left(\lambda_{\max}\right)$ at the corresponding band. The index *i* indicates the number of the corresponding band (*i* = 1, 2, 3).

The reflection area ($R\_Area_{i}$) was defined as:(6)$$R\_Area_{i}=\int_{\lambda_{j}}^{\lambda_{k}}(R_{i}\left(\lambda \right)-R_{ci}\left(\lambda \right))d\lambda ,$$
which is the definite integral of the difference between the reflectance $R_{i}\left(\lambda \right)$ in the corresponding band λ at the reflectance region and the inner continuum line $R_{ci}\left(\lambda \right)$. The wavelengths $\lambda_{j}$ and $\lambda_{k}$ are the initial and final wavelengths in each reflectance region, respectively. The index *i* is the number of the band (*i* = 1, 2, 3).

The normalized reflection depth ($R\_ND_{i}$) was the ratio of the reflectance depth ($R\_{\mathrm{Depth}}_{i}$) to the reflectance area ($R\_{\mathrm{Area}}_{i}$):(7)$$R\_ND_{i}=R\_{\mathrm{Depth}}_{i}/R\_{\mathrm{Area}}_{i}.$$

For more information, please see the relevant literature [19,20,21]. Table 2 showed the positional bands used in this study.

#### 2.2.3. Vegetation Indices

Many different optical indices have been reported in the literature and have proven to be well correlated with vegetation parameters. Consequently, we selected 34 vegetation indices (VIs) for estimating the corn canopy LNC (Table 3). Six of these indices were nitrogen-sensitive hyperspectral VIs [7], such as the optimal vegetation index (Vi_opt_), the normalized difference vegetation index green-blue^#^ (NDVI_g-b_^#^), the ratio vegetation index I^#^ (RVI I^#^), RVI II^#^, the combined index (MCARI/MTVI2), the double-peak canopy nitrogen index^#^ (DCNI^#^), NDVI I^#^, RVI III^#^, DVI I^#^, SAVI I^#^, normalized difference red-edge (NDRE), etc. Another 23 indices were used: The anti-atmospheric vegetation index (ARVI), the difference vegetation index (DVI II^#^), the enhanced vegetation index (EVI), the green normalized vegetation index (GNDVI), the modified nonlinear vegetation index (MNLI), the second modified SAVI (MSAVI2), the modified simple ratio (MSR), NDVI II^#^, the nonlinear vegetation index (NLI), the optimized SAVI (OSAVI), RDVI, the ratio vegetation index IV (RVI IV), SAVI II^#^, TVI and the modified triangle vegetation index (MTVI2), the red-edge-related index NDVI_Red-edge_, CI_Red-edge_, MTCI, the water-related index (WI, NDWI), the normalized difference infrared index (NDII), the water stress index (DSWI), the standardized LAI-determining index (sLAIDI*), etc. The VIs that related to the wide-band information were obtained from hyperspectral calculation using the spectral response functions of the corresponding sensors [22].

### 2.3. Screening and Modeling Methods

#### 2.3.1. Successive Projections Algorithm

In recent years, the successive projections algorithm (SPA) [23,24] has been ever more widely used for screening and extracting sensitive variables and is a forward-variable-selection algorithm that effectively eliminates the collinearity problem in spectral information. By reducing the redundancy between variables and selecting representative feature parameters for modeling, the efficiency of modeling analysis could be greatly improved.

In order to solve the collinearity problems, a minimally redundant subset of wavebands is selected in SPA and it belongs to the class of forward selection methods [57]. SPA starts with one wavelength and selects a new one at each iteration by using projection operators in a vector space until reaching the predefined number of wavelengths. Root means square error (RMSE) was used as the evaluation index. The final number of variables selected by the SPA is defined based on the lower RMSE value obtained.

#### 2.3.2. Partial Least Squares Regression

Partial least squares regression [58] is a statistical method that included principal component analysis, canonical correlation analysis, and multiple linear regression methods [59]. PLS regression is a modeling technique for studying multi-dependent variables or single-dependent variables and multi-independent variables. It can screen out low-collinearity components in the case of small sample size.

Consider *m* dependent variables $y_{1},y_{2},\ldots ,y_{m}$ and *n* arguments $x_{1},x_{2},\ldots ,x_{n}$. The quantities $E_{0},E_{1},\ldots ,E_{r},\;F_{0},\ldots ,F_{r}$ are standardized observation data arrays of two sets of variables from which we extract the components $t_{1},\ldots ,t_{r}\left(r\leq m\right)$,$t_{h}$ is a linear combination from the independent-variable set $X={\left(x_{1},\ldots ,x_{m}\right)}^{T}$ and carries the maximum information possible from X. At the same time, $t_{h}$ has the greatest explanatory power for the dependent-variable system $F_{0}$. If we extract *r* components $t_{1},\ldots ,t_{r}$ from the independent-variable set, the PLS regression will establish the regression equation for $y_{1},y_{2},\ldots ,y_{m}$ and $t_{1},\ldots ,t_{r}$, and then express $y_{1},y_{2},\ldots ,y_{m}$ and the regression equation of the original independent variables; that is, the PLS regression equation:(8)$$y_{j}=\;a_{j1}x_{1}+\ldots +a_{jm}x_{m},\left(j=1,2,\;\ldots ,\;p\right).$$

#### 2.3.3. Random Forest

Random forest (RF) [60,61] used bootstrapping to randomly draw samples that were resampled and put back. The extracted samples are used to construct a classification decision tree, and the non-extracted samples constitute the out-of-bag (OOB) data set.

Given *n* features, RF arbitrarily extracts less than *m* (*m* < *n*) features at each node of each tree, selects the classification of the decision tree with the largest amount of information among the *m* features, and does not prune the classification decision tree.

A plurality of regression decision trees is constructed by using the extracted samples to form a RF, and then the data are classified, and the result is decided by voting.

Each time a RF forms, the OOB data set is used to evaluate the classification results, following which we evaluate the combined classifier. The variable that generates the decision tree is randomly selected each time from the training set, so the random forest has a stable error rate, and each OOB generated can be used to evaluate the classifier performance.

#### 2.3.4. Statistical Analysis Method

Between the sensitive bands, the location characteristics, VIs, and corn canopy LNC were analyzed by using Rstudio 3.5.3. The validation samples were one-third of the samples (i.e., 24 samples) and did not participate in the validation. The operation of partial least squares and random forest algorithm was done in MATLAB R2014a. The determination coefficient *R*^2^, the RMSE, and the normalized root mean square error (NRMSE) serve as indicators to explain and quantify the relationship with nitrogen in canopy leaves. They were calculated as follows:(9)$$R^{2}={\left(\Sigma_{i=1}^{n}y_{i}-\bar{y}\right)}^{2}/{\left(\Sigma_{i=1}^{n}x_{i}-\bar{y}\right)}^{2},$$
(10)$$\mathrm{RMSE}=\sqrt{\Sigma_{i=1,j=1}^{n}{\left(x_{i}-y_{j}\right)}^{2}/n\;},$$
(11)$$\mathrm{NRMSE}=\sqrt{\Sigma_{i=1,j=1}^{n}{\left(x_{i}-y_{j}\right)}^{2}/n}/\bar{y},$$
where $x_{i}$ is the measured nitrogen content in corn canopy leaves, $y_{i}$ is the predicted nitrogen content in corn canopy leaves, $\bar{y}$ is the mean nitrogen content in corn canopy leaves, and n is the number of samples.

## 3. Results

### 3.1. Optimal Spectral Characteristics

#### 3.1.1. Sensitive Reflectance Feature Data Set

Figure 3 shows a correlation between the corn canopy reflectance spectra and the FD spectra and LNC. The reflectance spectrum in Figure 3 suggests a negative correlation between the blue (630 nm), the red (711 nm), and the short-wave near-infrared (1996–2346 nm) reflectance spectra, and a positive correlation in the near-infrared (739–1135 nm). The FD spectrum shows a positive correlation at 661 and 751 nm, and a negative correlation at 691 nm. Spectral first derivative (FD) operation reduces the background noise and raises the efficiency of the target. The correlation coefficient of the FD spectrum is greater than the reflection spectrum in the visible light and NIR range (400–1400 nm). Using the SPA algorithm, four reflectance wavelengths and two FD wavelengths were selected to form a sensitive spectral dataset (Figure 4). The 724, 1343 nm (Ref), 658, and 937 nm (FD) wavelengths were well correlated with LNC, and 724 nm, 658 nm, and 937 nm fell in the visible light and NIR bands, indicating that the corn canopy reflectance spectrum and the FD spectrum were strongly correlated with LNC in the visible range (400–700 nm) and NIR range (700–800 nm).

The original spectrum is easily affected by illumination, soil background, atmosphere, and other factors. However, derivative transformation can reduce or eliminate the influence of background and atmospheric scattering and improve the contrast of different absorption characteristics. Therefore the reflection spectral bands and the first derivative bands selected were used separately in PLS and RF models.

The SPA screened out six sensitive spectra (including four reflectance spectral wavelengths: 412, 724, 1084, and 1343 nm and two first derivative spectral wavelengths: 658 and 937 nm) with the least linearity of leaf nitrogen content. A model was established between the four reflectance spectra and the LNC based on the PLS and RF regression. Similarly, a model was established between the two first derivative spectra and LNC based on the PLS and RF regression. The results are given in Table 4. The PLS model was used to estimate the nitrogen content of leaves. The coefficient *R*^2^, the RMSE, and the NRMSE of the reflection spectral bands were 0.59, 38.2%, and 0.20, respectively. For the FD bands, these values are 0.54, 39.7%, and 0.21, respectively. The RF model was used to estimate the nitrogen content of leaves. The coefficient R^2^, the RMSE, and the NRMSE of the bands of the reflection spectrum were 0.61, 42.1%, and 0.22, respectively; and for the FD bands these values were 0.59, 37.9%, and 0.20, respectively. These values represented good results for the modeling.

In the validation set, when PLS was used to estimate the LNC, *R*^2^ for the reflectance spectrum was 0.22 greater than for the RF model, the RMSE was 0.2 less, and the NRMSE was 11.2% less. The coefficient *R*^2^ of the RF model for the FD value was 0.02 less than the Ref, the RMSE was 0.05 less, and the NRMSE was 2.6% less. These results showed that the inversion of the corn LNC by the PLS model was better than the RF model for the reflectance spectra, which suggested that the PLS model should provide more accurate predictions of the LNC. However, the RF model was more stable than the PLS model.

#### 3.1.2. Position Feature Data Set

We selected the position characteristics of 40 hyperspectral reflectance wavelengths and calculated the positional correlation of each calibration set (75%; Figure 5). The results showed that the LNC was strongly correlated with Db, Dr, λb, Rg, λg, and SDb, whereas the LNC was weakly correlated with the other parameters. Two positional parameters SDb and Dr with smaller collinearity were selected by the SPA algorithm and were modeled using the LNC (Table 5). The results of estimating the LNC in the optical layer for *R*^2^, RMSE, and NRMSE were 0.50, 41.2%, 0.22 and 0.57, 39.9%, 0.21 for the PLS and RF models, respectively. The coefficient *R*^2^ of the RF model was 0.07 greater, the RMSE was 0.01 less, and the NRMSE was 0.7% less than for the PLS model, which indicated that the RF model was more stable and the PLS model had better results than the RF model. The coefficient *R*^2^ of the PLS model in the validation set was 0.1 greater, the RMSE was 0.07 lower, and the NRMSE was 3.2% lower than the RF model.

#### 3.1.3. Vegetation Indices Data Set

For this study, 34 vegetation indices (VIs; Figure 6) were selected to study the correlation between them and LNC. The results show that DVI I, DVI II, and TVI had a weaker relevance with LNC than others. Such results are expressed in Figure 6, where these VIs are represented by smaller circles and lighter color when compared to the other VIs.

The SPA algorithm selected the eigenvalues NDVI_g-b_^#^ and DVI II, which had small collinearity between VIs. The two parameters were modeled with the LNC (Table 6), and the results show that *R*^2^, RMSE, and NRMSE were 0.68, 33.3%, 0.17 and 0.64, 35.6%, 0.19, when the canopy LNC was estimated by the PLS model and the RF model, respectively. For the estimation, the coefficient *R*^2^ was 0.04 greater, the RMSE was 0.03 less, and the NRMSE was 0.8% less for the PLS model than for the RF model. For the validation model, the coefficient *R*^2^ was 0.2 greater, the RMSE was 11.2% less, and the NRMSE was 0.06 less for the PLS model than for the RF model. These results indicated that the RF model was more stable and that the PLS model might provide more accurate results to estimate the LNC.

### 3.2. Composite Spectral Features

To further improve the accuracy of the spectral estimates of the corn canopy LNC, six sensitive spectral features (four reflective spectral features, two FD features), two positional features, and two VIs were obtained from the three spectral variables. The SPA algorithm was then used to further screen the sensitive characteristic parameters to model, which were combined into a new set of spectral variables. The analysis shows that the spectral reflection bands at 724 and 1343 nm, FD band at 658 nm, and NDVI_g-b_^#^ became the new sensitive spectral variables. Table 7 gives the results of using these new characteristic parameters to estimate the corn canopy LNC. For estimating the corn canopy LNC, *R*^2^ was 0.14 greater, RMSE was 7.5% lower, and the NRMSE was 0.03 lower for the PLS model than for the RF model. Figure 7a,b show the results of the validation model. The fit was better between the measured value and the predicted value, *R*^2^ was 0.22 greater, RMSE was 0.12 less, and NRMSE was 6.8% less for the PLS model than for the RF model. The results of the RF model did not differ significantly, and the results were relatively stable. However, the result of the PLS model was better.

## 4. Discussion

In order to reduce the influence of water vapor and other factors of hyperspectral data [62,63], we chose 400–1353 nm, 1437–1799 nm, and 1992–2354 nm to study the spectra. We selected reflectance spectra of 412, 724, 1084, and 1343 nm and first derivative spectra of 658 and 937 nm. Kokaly and Clark [18] got the spectral characteristics of absorption and reflection positions via using the continuum-removal method. We selected two positions characteristic using the same method: SDb and Dr. The vegetation indices were a linear and non-linear combination of different bands, and the functional relationship of vegetation characteristic parameters was more stable and reliable than a single band [64]. We selected NDVI_g-b_^#^, DVI II of the two optimal VIs.

Serious multi-collinearity problems arose in sensitive bands, positions and VIs. The optimal sensitive band features, position features, and VIs of hyperspectral data were selected by using SPA and they had a good correlation with LNC (Figure 4, Figure 5 and Figure 6), but the correlation between the position features and LNC was low. Bands that the optimal parameters used were mainly focused on visible- and near-infrared-band. This result was consistent with the results of a previous study [65]. In this paper, the *R*^2^ between the optimal reflectance spectra, VIs and LNC achieved 0.82 and 0.80. The parameters were mainly concentrated on blue-light, red-edge, and NIR bands, it might be the influence of internal factors such as chlorophyll and cell of plants, and the position parameters were red-shifted due to the difference of leaves nitrogen content. The integrated spectral features (reflectance at 724 and 1343 nm, FD at 658 nm, and NDVIg-b^#^) determined *R*^2^, the RMSE, and the NRMSE for the calibration set (validation set) of the PLS model and the RF model to be 0.71, 31.8%, 0.17 (0.77, 31.0%, 0.17) and 0.57, 39.3%, and 0.20 (0.55, 43.3%, and 0.24), respectively. Chen et al. [7] suggested that the *R*^2^ values were 0.72 for corn. Ours results were increased by 0.05 than theirs in the PLS model and there were no significant differences in values. The composite spectral features integrated characteristics of three variable sets and the results of PLS model were more stable, when comparing the results for calibration and validation datasets, than any other three variable datasets used independently. The reflectance bands and position features were easily affected by external light, water and nitrogen content, and so on. VIs had the ability to eliminate effects of soil background factors, especially NDVIg-b^#^.

Most previous studies focused on a single variable of the spectrum [2,11,66] to study corn leaves nitrogen content, whereas few studies discussed the comprehensive processing of data or compared models with similar variables. The present study used two models for the analysis: The PLS and RF models. The LNC model established by the PLS algorithm was the best in the two models. The method of linear model was obviously better than machine learning. The results of the PLS model could decompose and filter the data by leveraging the number of input samples. A high precision model was established for the comprehensive variable with the strongest explanatory power of the dependent variable [67,68]. The RF model is a machine learning algorithm with simple implementation, good precision, and strong over-fitting ability [69]. This is indicative of a strong learning ability, which is consistent with the results of Feng et al. In the process of modeling with the four variable datasets as the independent variable and LNC as dependent variables, the modeling result of the RF model was not very different, but the result of the PLS model was better, which could better predict LNC. Based on sensitive variables obtained from screened multi-variety and multi-growth data over one year, the next step is to lengthen the study (multiple years) and use more regional data for an in-depth analysis.

## 5. Conclusions

In this paper, the results showed that the spectral bands, absorption and reflection positions, and VIs were usually good predictors. LNC had better correlation with the optimal sensitive bands and VIs (Figure 4 and Figure 5), but the optimal positions have a bad correlation with LNC (Figure 6). These hyperspectral features were mostly concentrated on the 300–1400 nm region, and the features in visible light and NIR regions was able to better realize the monitoring of corn LNC [70].

After screening out the original spectral bands for sensitive reflect feature dataset by SPA, for the validation set, the R^2^, RMSE, and NRMSE of the PLS model and RF model were 0.82, 27.5%, and 0.15 and 0.64, 37.9%, and 0.21, respectively. The R^2^, RMSE, and NRMSE for the first derivative bands of the PLS model and RF model were 0.60, 47.7%, and 0.26 and 0.58, 42.8%, and 0.24, respectively. The R^2^, RMSE, and NRMSE for the position feature dataset of the PLS model and RF model were 0.62, 41.5%, and 0.23 and 0.52, 47.2%, and 0.26, respectively. The R^2^, RMSE, and NRMSE for the vegetation indices feature dataset of the PLS model and RF model were 0.80, 0.31, and 16.9% and 0.60, 0.42, and 23.1%, respectively. The R^2^, RMSE, and NRMSE for the reflect feature integration dataset of the PLS model and RF model were 0.77, 0.31, and 17.1% and 0.55, 0.43, and 23.9%, respectively. For estimating the corn LNC, the RF model had a good learning ability and stable results. However, the results of *R*^2^, RMSE, and NRMSE were poor in the validation set with a small sample size, while the results of PLS were good, especially in the integration dataset, which could better estimate the LNC.

## Figures and Tables

**Figure 1 sensors-19-02898-f001:**
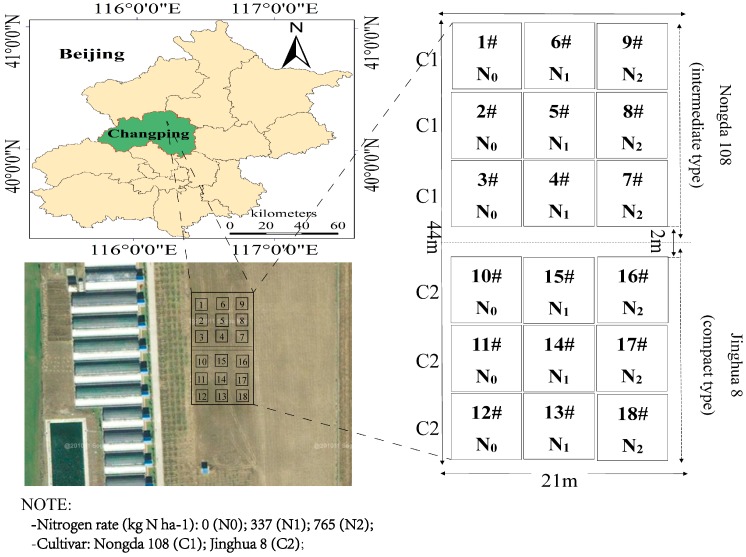
Study plots.

**Figure 2 sensors-19-02898-f002:**
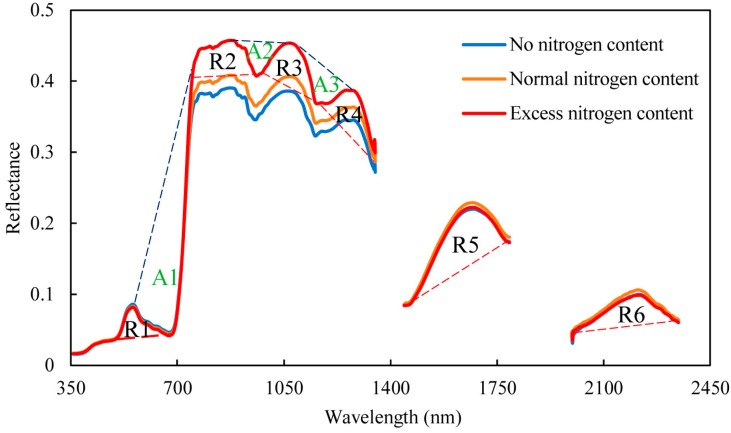
Characteristic (A) absorption and (R) reflection positions of the summer corn for the three nitrogen treatments. The red dotted line was the inner continuous line $R_{ci}\left(\lambda_{\max}\right)$ in the reflection position, the blue dotted line was outer continuum line $R_{ci}\left(\lambda_{\min}\right)$ in the absorption region.

**Figure 3 sensors-19-02898-f003:**
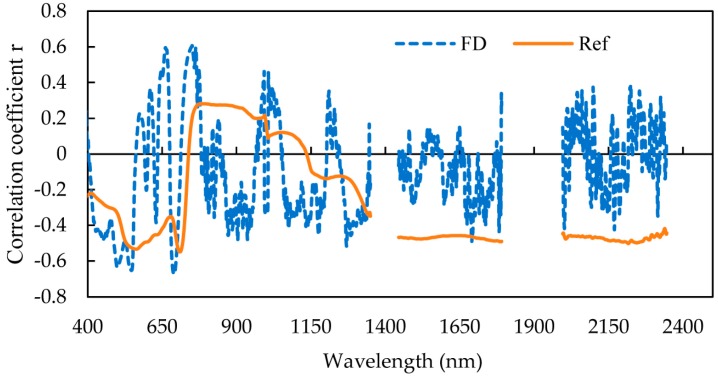
Correlation between model set reflectance spectra (Ref) and first derivative spectra (FD) and LNC of the training set.

**Figure 4 sensors-19-02898-f004:**
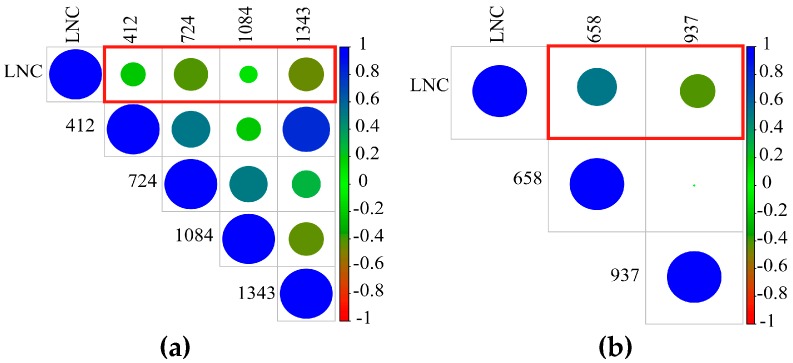
Correlation between selected sensitive spectral bands and LNC (in red boxes, *n* = 48). (**a**) 412, 724, 1084, and 1343 nm are reflectance spectra; and (**b**) 658 and 937 nm are first derivative spectra.

**Figure 5 sensors-19-02898-f005:**
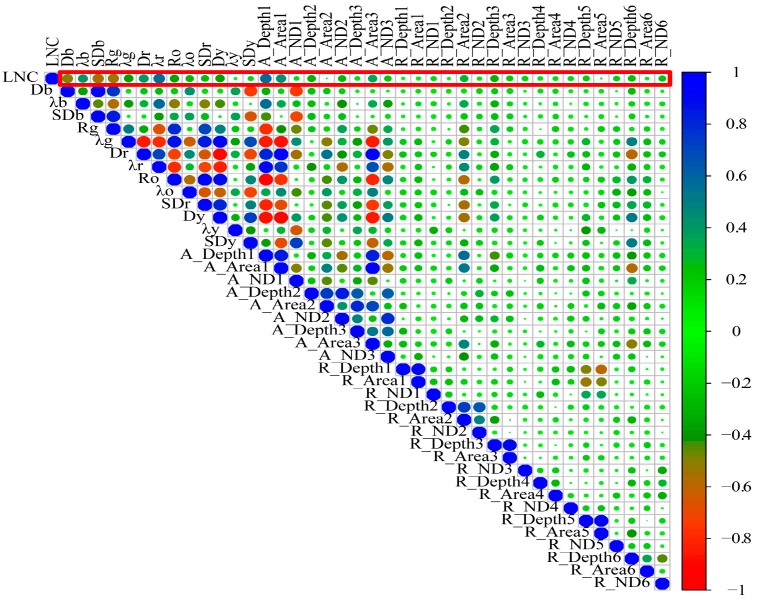
Correlations between LNC and special positions (in red boxes, *n* = 48).

**Figure 6 sensors-19-02898-f006:**
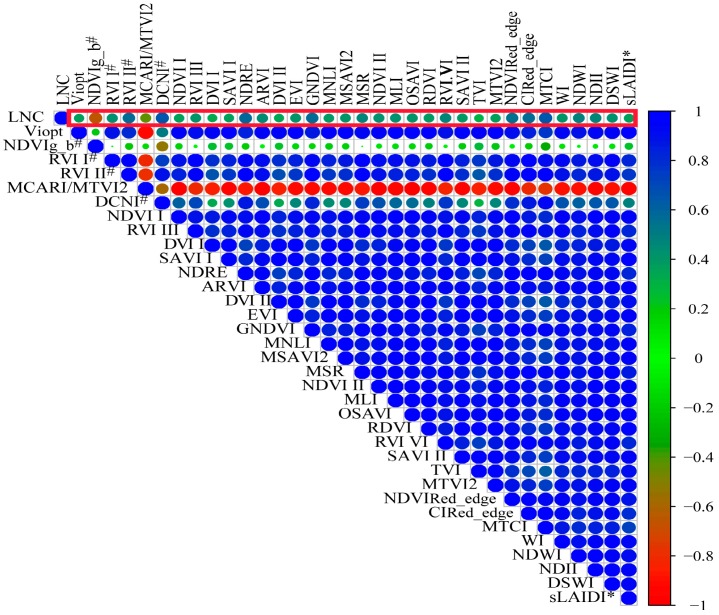
Correlation and significance of vegetation indices (VIs) and LNC (in red boxes, *n* = 48).

**Figure 7 sensors-19-02898-f007:**
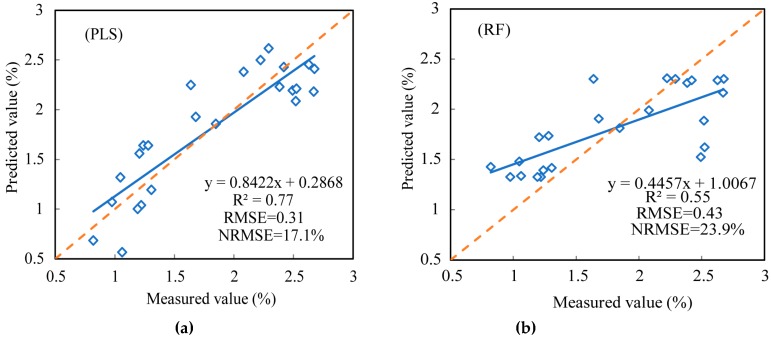
Accuracy of the measured and predicted values of the validation set: (**a**) Partial least squares (PLS) model, (**b**) random forest (RF) model.

**Table 1 sensors-19-02898-t001:** Descriptive statistics of leaf nitrogen content (LNC).

Dataset	Year	Samples	Max	Min	Mean	SD	Coefficient of Variation
Calibration dataset	2012	48	2.83	0.92	1.91	0.59	0.31
Validation dataset	2012	24	2.68	0.82	1.81	0.65	0.36

**Table 2 sensors-19-02898-t002:** Partial characteristic variables of the hyperspectral data.

Variables	Definition and Description
Db	Maximum value of the 1st derivative with a blue edge (490–530 nm)
λb	Wavelength at Db
Dy	Maximum value of the 1st derivative with a yellow edge (560–640 nm)
λy	Wavelength at Dy
Dr	Maximum value of the 1st derivative with a red edge (680–760 nm)
λr	Wavelength at Dr
Rg	Maximum reflectance with a green peak (510–560 nm)
λg	Wavelength at Rg
Ro	Lowest reflectance with a red well (650–690 nm)
λo	Wavelength at Ro
SDb	Sum of the 1st derivative values within the blue edge
SDy	Sum of the 1st derivative values within the yellow edge
SDr	Sum of the 1st derivative values within the red well

**Table 3 sensors-19-02898-t003:** Summary of partial vegetation indices.

Index	Name	Formula	Reference
Vi_opt_	Optimal vegetation index	(1 + 0.45) ((R_800_)^2^ + 1)/(R_670_ + 0.45)	[25]
NDVI_g-b_^#^	Normalized difference vegetation index^#^	(R_573_ − R_440_)/(R_573_ + R_440_)	[26]
RVI I^#^	Ratio vegetation index I^#^	R_810_/R_660_	[27]
RVI II^#^	Ratio vegetation index II^#^	R_810_/R_560_	[28]
MCARI/MTVI2	Combined index	MCARI/MTVI2 MCARI:(R_700_ − R_670_ − 0.2(R_700_ − R_550_)) (R_700_/R_670_) MTVI2: $1.5\left(1.2\left(R_{800}-R_{550}\right)\right)/\mathrm{sqrt}\left(\left(2R_{800}+1\right)2-\left(6R_{800}-5\mathrm{sqrt}\left(R_{670}\right)\right)-0.5\right)$	[29]
DCNI^#^	Double-peak canopy nitrogen index I^#^	(R_720_ − R_700_)/(R_700_ − R_670_)/(R_720_ − R_670_ + 0.03)	[7]
NDVI I	Normalized difference vegetation index I	(R_800_ − R_670_)/(R_800_ + R_670_)	[30]
RVI III	Ratio vegetation index III	R_800_/R_670_	[31]
DVI I	Difference vegetation index I	R_800_-R_670_	[32]
SAVI I	Soil-adjusted vegetation index I	1.5(R_800_ − R_670_)/(R_800_ + R_670_ + 0.5)	[33]
NDRE	Normalized difference red edge	(R_790_ − R_720_)/(R_790_ + R_720_)	[34]
ARVI	Atmospherically-resistant vegetation index	ARVI = (R_NIR_ − RB)/(R_NIR_ + RB) RB = R-γ(B-R), γ = 1	[35]
DVI II	Difference vegetation index II	DVI = R_NIR_ − R_R_	[36]
EVI	Enhanced vegetation index	EVI = 2.5(R_NIR_ − R_R_)/(R_NIR_ + 6R_R_ − 7.5R_B_ + 1)	[37]
GNDVI	Green normalized difference vegetation index	GNDVI = (R_NIR_ − R_R_)/(R_NIR_ + R_R_)	[38]
MNLI	Modified nonlinear vegetation index	MNLI = 1.5(R_NIR_^2^ − R_R_)/(R_NIR_^2^ + R_R_ + 0.5)	[39]
MSAVI2	The second modified SAVI	MSAVI2 = $\left(2R_{\mathrm{NIR}}+1-\mathrm{sqrt}\left({\left(2R_{\mathrm{NIR}}+1\right)}^{2}-8\left(R_{\mathrm{NIR}}-R_{R}\right)\right)\right)/2$	[40]
MSR	Modified simple ratio	MSR = (R_NIR_/R_R_ − 1) / (R_NIR_/R_R_ + 1)	[41]
NDVI II	Normalized difference vegetation index II	NDVI = (R_NIR_ − R_R_) / (R_NIR_ + R_R_)	[42]
NLI	Nonlinear vegetation index	NLI = (R_NIR_^2^ − R_R_)/(R_NIR_^2^ + R_R_)	[43]
OSAVI	Optimization of soil-adjusted vegetation index	OSAVI = (1 + 0.16) (R_NIR_ − R_R_)/(R_NIR_ + R_R_ + 0.16)	[44]
RDVI	Renormalization difference vegetation index	RDVI = $\left(R_{\mathrm{NIR}}-R_{R}\right)/\left(\mathrm{sqrt}\left(R_{\mathrm{NIR}}+R_{R}\right)\right)$	[45]
RVI IV	Ratio vegetation index	RVI = R_NIR_/R_R_	[33]
SAVI II	Soil-adjusted vegetation index II	SAVI = 1.5(R_NIR_ − R_R_)/ (R_NIR_ + R_R_ + 0.5)	[46]
TVI	Triangular vegetation index	TVI = 60(R_NIR_ − R_G_) − 100(R_R_ − R_G_)	[47]
MTVI2	Modified triangular vegetation index	MTVI2 = 1.5(1.2(R_NIR_ − R_G_) − 2.5(R_R_ − R_G_))/(sqrt ((2R_NIR_ + 1)^2^ − (6R_NIR_ − 5sqrt (R_R_) − 0.5))	[48]
NDVI_Red-edge_	Red-edge NDVI	NDVI_Red-edge_ = (R_NIR_ − R_Red-edge_)/(R_NIR_ − R_Red-edge_)	[49]
CI_Red-edge_	Red-edge Chlorophyll Index	CI_Red-edge_ = (R_NIR_/R_Red-edge_) − 1	[50]
MTCI	MERIS Terrestrial Chlorophyll Index	MTCI = (R_NIR_ − R_Red-edge_)/(R_Red-edge_ − R_NIR_)	[51]
WI	Water Index	WI = R_900_/R_970_	[52]
NDWI	Normalized difference water index	NDWI = (R_860_ − R_1240_)/(R_860_ + R_1240_)	[53]
NDII	Normalized difference infrared index	NDII = (R_819_ − R_1600_)/(R_819_ + R_1600_)	[54]
DSWI	Disease water stress index	DSWI = (R_803_ − R_549_)/(R_1659_ + R_681_)	[55]
sLAIDI *	Standardized LAI-determining index	sLAIDI * = s(R_1050_ − R_1250_)/(R_1050_ + R_1250_)R_1555_, s = 1	[56]

* I, II, III, IV, V were just only for the same planting indices that distinguished different bands.

**Table 4 sensors-19-02898-t004:** Accuracy between the sensitive band and FD spectrum and LNC.

Algorithm	Feature Types	Calibration Set (*n* = 48)			Validation Set (*n* = 24)		
		R^2^	RMSE	NRMSE	R^2^	RMSE	NRMSE
Partial Least Squares (PLS)	Ref	0.59	0.38	19.8%	0.82	0.28	15.2%
	FD	0.54	0.40	20.8%	0.64	0.38	21.0%
Random Forest (RF)	Ref	0.61	0.42	22.1%	0.60	0.48	26.4%
	FD	0.59	0.38	19.9%	0.58	0.43	23.6%

**Table 5 sensors-19-02898-t005:** Modeling results between position features and LNC.

Algorithm	Feature Types	Calibration set (*n* = 48)			Validation set (*n* = 24)		
		R^2^	RMSE	NRMSE	R^2^	RMSE	NRMSE
Partial Least Squares (PLS)	Positions	0.50	0.41	21.6%	0.62	0.41	22.9%
Random Forest (RF)	Positions	0.57	0.40	20.9%	0.52	0.47	26.1%

**Table 6 sensors-19-02898-t006:** Modeling results between VIs and LNC.

Algorithm	Feature Types	Calibration Set (*n* = 48)			Validation Set (*n* = 24)		
		R^2^	RMSE	NRMSE	R^2^	RMSE	NRMSE
Partial Least Squares (PLS)	VIs	0.68	0.33	17.4%	0.80	0.31	16.9%
Random Forest (RF)	VIs	0.64	0.36	18.6%	0.60	0.42	23.1%

**Table 7 sensors-19-02898-t007:** Modeling results between comprehensive parameters and LNC.

Algorithm	Feature Types	Calibration Set (*n* = 48)			Validation Set (*n* = 24)		
		R^2^	RMSE	NRMSE	R^2^	RMSE	NRMSE
Partial Least Squares (PLS)	Integrated data	0.71	0.32	16.7%	0.77	0.31	17.1%
Random Forest (RF)	Integrated data	0.57	0.39	20.4%	0.55	0.43	23.9%

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
