# Peer review of "Hyperspectral-Based Estimation of Leaf Nitrogen Content in Corn Using Optimal Selection of Multiple Spectral Variables"

_sensors, 2019, doi:10.3390/s19132898_

Round 1
Reviewer 1 Report
The manuscript presents a methodology for estimating leaf nitrogen content (LNC) using hyperspectral data obtained through a handheld spectroradiometer.
In my opinion, the topic of the manuscript is marginal to the scope of Sensors journal. There is nothing new regarding the sensor used. Also, the use of hyperspectral data derived from handheld spectroradiometers for estimating LNC is not specifically new. As such, there is no novelty regarding the sensing system used.
English language and style needs to be improved throughout the manuscript. Overall, the manuscript content needs improvements and a rigorous presentation and interpretation of information is required.
Introduction section needs an improvement of content presentation. For example, lines 48 – 77 need to be better focused regarding the characteristics of each spectral variables type but also regarding the applications being used as example. Also, it is relevant including up to date information as well as information from studies published in journals with impact factor.
In the section 3, the results are presented according to each spectral variables type: (i) sensitive spectral feature; (ii) position feature dataset; (iii) vegetation indices; and (iv) composite spectral features. Regarding the sensitive spectral features, 4 reflectance wavelengths and 2 first derivative wavelengths were selected. It’s not clear to me why the reflection spectral bands and the first derivative bands selected were used separately in PLS and RF models. Regarding the vegetation indices (section 3.1.3), the first paragraph of the section (lines 320 – 323) is very confusing.
The discussion and conclusions sections need to be significantly improved. A deeper discussion of the various results presented is required. No discussion is provided about the advantages and the disadvantages of the different types of inputs (spectral variable types)used in the models. Somehow, the discussion and conclusion don’t seem to be consistent with the results. The Authors advocate that using composite spectral features obtained better performance of the PLS and RF models, which is also indicated in the abstract (lines 26-30). However, when using only the two selected vegetation indices the results of validation were better (with less input variables) than the results with composite spectral features. Also, in the conclusion section, lines 401 – 407 are presented values that are not consistent with the results presented in the tables of results section.
Author Response
Manuscript ID: sensors-519557
Title: Hyperspectral-based Estimation of Nitrogen Content in Corn Leaves Using Optimal Selection of Multiple Spectral VariablesAuthor: Lingling Fan, Jinling Zhao*, Xingang Xu, Dong Liang, Guijun Yang, Haikuan Feng, Hao Yang, Yulong Wang, Guo Chen, Penfei Wei
Original Research
Scientific Reports
Cover letter
Dear reviewer,
Thank you very much for your reply and reviewers’ constructive suggestions for improving our manuscript (sensors-519557). We revised the manuscript point by point very carefully according to the reviewers’ suggestions. Attached please find the revised manuscript and the following responses to reviewers’ comments. In the manuscript, all revised parts are highlighted in red for convenience of being reviewed by you and reviewers.
With best regards,
Lingling Fan and co-authors
****************************************************
Correspondence information: Jinling Zhao
National Engineering Research Center for Agro-Ecological Big Data Analysis & Application, Anhui University, Hefei 230601, China.
E-mail: [email protected]
Telephone: +(0551) 6295 0280
****************************************************
Response to Reviewer 1 Comments
Point 1、In my opinion, the topic of the manuscript is marginal to the scope of Sensors journal. There is nothing new regarding the sensor used. Also, the use of hyperspectral data derived from handheld spectroradiometers for estimating LNC is not specifically new. As such, there is no novelty regarding the sensing system used.
Response:
1) The writing uses the data collected by handheld spectroradiometer, which lays a foundation for the subsequent development of this paper. This is very important.
2) There are few studies on comprehensive and systematic analysis of nitrogen content in corn leaves from three aspects. From the perspectives of spectral reflectance, positions and vegetation indices, spectral variables are selected by SPA algorithm and analyzed by PLS and RF models with LNC. Compared with the other three variable sets, the results are similar and stable of the PLS model in integration variable dataset. I hope that my research of these aspects can provide readers with some reference.
Point 2、English language and style needs to be improved throughout the manuscript. Overall, the manuscript content needs improvements and a rigorous presentation and interpretation of information is required.
Response: We have invited an expert who has a good command of English to revise the language problems of the manuscript and I also revise the content. Thank for your good suggestion. (The revised details are corresponding to the PDF file from the reviewer)
Point 3、Introduction section needs an improvement of content presentation. For example, lines 48 – 77 need to be better focused regarding the characteristics of each spectral variables type but also regarding the applications being used as example. Also, it is relevant including up to date information as well as information from studies published in journals with impact factor.
Response: Thank for your good suggestion. I have made the following adjustments and added content. (Details are highlighted in Introduction (Line 8-57) of the revised manuscript)
Point 4、In the section 3, the results are presented according to each spectral variables type: (i) sensitive spectral feature; (ii) position feature dataset; (iii) vegetation indices; and (iv) composite spectral features. Regarding the sensitive spectral features, 4 reflectance wavelengths and 2 first derivative wavelengths were selected. It’s not clear to me why the reflection spectral bands and the first derivative bands selected were used separately in PLS and RF models. Regarding the vegetation indices (section 3.1.3), the first paragraph of the section (lines 320 – 323) is very confusing.
Response: I'm sorry for the trouble I've caused you.
1)The original spectrum is easily affected by illumination, soil background, atmosphere and other factors. However, derivative transformation can reduce or eliminate the influence of background and atmospheric scattering and improve the contrast of different absorption characteristics. So the reflection spectral bands and the first derivative bands selected were used separately in PLS and RF models.
2)Regarding the vegetation indices (section 3.1.3), I have made the following adjustments. (Details are highlighted in 3.1.3. Vegetation indices data set (Line 300-303) of the revised manuscript)
Point 5、The discussion and conclusions sections need to be significantly improved. A deeper discussion of the various results presented is required. No discussion is provided about the advantages and the disadvantages of the different types of inputs (spectral variable types) used in the models. Somehow, the discussion and conclusion don’t seem to be consistent with the results. The Authors advocate that using composite spectral features obtained better performance of the PLS and RF models, which is also indicated in the abstract (lines 26-30). However, when using only the two selected vegetation indices the results of validation were better (with less input variables) than the results with composite spectral features. Also, in the conclusion section, lines 401 – 407 are presented values that are not consistent with the results presented in the tables of results section.
Response: Thank you for your suggestions.
1) I've improved the discussion and conclusion and revised the results in the conclusion section. (Details are highlighted in 4. Discussion (Line 338-366) and 5. Conclusion (Line 382-395) of the revised manuscript)
2) Although using only the two selected vegetation indices the results of validation were better (with less input variables) than the results with composite spectral features, The R2, RMSE and NRMSE for reflect feature integration dataset of the PLS model are 0.71, 0.32, 16.7% (calibration set) and 0.77, 0.31, 17.1% (validation set) , the results are not much different and more stable. What’s more, the integrated variable set contains the optimal features of reflectance spectra set, position set and vegetation indices set, and it can eliminate the influence of internal data instability to a certain extent. (Details are highlighted in Table 7 (Line 334-335) of the revised manuscript)

Reviewer 2 Report
This paper presents an interesting approach to the problem of estimating vegetation properties (leaf nitrogen content in this case) from spectral characteristics. Using statistical methods the authors isolate some spectral features that exhibit a good correlation with leaf nitrogen content.
However, the experimental procedure is not very well described and the statistical analysis is sometimes diffictult to follow.
Furthermore, the manuscript uses verbal tenses in a weird way. I think the paper needs to be take care of this.
I think the authors should address the problems indicated below before accepting this paper for publication:
1) Line 104: pH of the soil is included in a list of nutrients. Is this correct? I am afraid pH should be given as a property of the soil apart of the nutrient list.
2) Lines 113-114: what do V6, V14, R1 and R2 stand for? They are not used anymore throughout the manuscript.
3) Table 2.- What does C.V. stand for? It must be explained somewhere.
4) Lines 131-132: the authors say that they used data from 72 experimental plots. They have t explain how many of these plots correspond to each corn species, and how many of these plots correspond to the V6, V14, R1 and R2 seasons. They also have to explain the extension of each of the plots, and how they were chosen. I think this part of the experiment has to be explained in more detail.
5) Regarding Spectral position features: how do the authors calculate the base lines to calculate the absorption depth and the reflection depth?
6) Reganding 10) Spectral position features: in fig. 2, I guess the dashed lines are the base lines used to calculate the absorption depth and the reflection depth. Please, in the figure caption the meaning of the dashed lines should be given.
7) Figure 2. The caption of this figure includes some text in Chinese that should be translated into English or deleted.
8) Line 217-218: The authors state: “The number with minimum RMSE was the number of variables”: I do not understand the meaning of this sentence. What does this have to do with how SPA works?
10) Line 284-285: the authors state that “The SPA algorithm screened out six sensitive reflection spectra with the least linearity of leaf nitrogen content, and six sensitive spectra were modeled by PLS and RF regression”. Do they really mean six sensitive reflection spectra? I do not understand how SPA is used, I thought it was being used to screen out spectral characteristics, not spectra. Furthermore, if only six spectra were screened out to be used with PLS and RF, why is it that they used 48 spectra for calibration and 24 for validation, as shown in Table 5. This is not clear at all.
11) Another point is that if SPA is used to screen out spectral characteristics, the author must give the six spectral characterisctics that were used in PLS and RF.
12) Line 363-364: The authors state that: “Applying the SPA algorithm was selected three optimal sensitive variables of hyperspectral data”. But on Line 340 they state that 4 spectral characteristics were selected using SPA : “The analysis showed that the spectral reflection bands at 724 and 1343 nm, FD 340 band at 658 nm, and NDVIg-b” This makes 4 spectral charactersitics. My question is: how many spectral characteristics were chosen: three or four?
13) As for the English language, there are a lot of sentences that are not correctly written, and verbal tenses are not correctly used. Some examples:
Line 189: edit
Line 190: “summarized”
Line 212: “had been became”
Line 213: “SPA was”
Line 220: “was”, “combined”
Line 264: “revealed”
Lien 265: “showed”
Line 286: “were”
Line 336: “improved”
Line 358-359: ·”were a poor correlation”
Linde 363-364: Applying the SPA algorithm was selected three optimal sensitive variables of hyperspectral data”
Author Response
Manuscript ID: sensors-519557
Title: Hyperspectral-based Estimation of Nitrogen Content in Corn Leaves Using
Optimal Selection of Multiple Spectral Variables
Author: Lingling Fan, Jinling Zhao*, Xingang Xu, Dong Liang, Guijun Yang, Haikuan
Feng, Hao Yang, Yulong Wang, Guo Chen, Penfei Wei
Original Research
Scientific Reports
Cover letter
Dear reviewer,
Thank you very much for your reply and reviewers’ constructive suggestions for
improving our manuscript (sensors-519557). We revised the manuscript point by point
very carefully according to the reviewers’ suggestions. Attached please find the revised
manuscript and the following responses to reviewers’ comments. In the manuscript, all
revised parts are highlighted in red for convenience of being reviewed by you and
reviewers.
With best regards,
Lingling Fan and co-authors
****************************************************
Correspondence information: Jinling Zhao
National Engineering Research Center for Agro-Ecological Big Data Analysis &
Application, Anhui University, Hefei 230601, China.
E-mail: [email protected]
Telephone: +(0551) 6295 0280
****************************************************
Response to Reviewer 2 Comments
Point 1、Line 104: pH of the soil is included in a list of nutrients. Is this correct? I am
afraid pH should be given as a property of the soil apart of the nutrient list.
Response: Thank you for your good suggestion. In order to describe the soil apart of
the nutrient, I add the content of the pH . (Details are highlighted in 2.1.1. Study area
(Line 73-74) of the revised manuscript)
Point 2、Lines 113-114: what do V6, V14, R1 and R2 stand for? They are not used
anymore throughout the manuscript.
Response: V6 refers to 6 leaf stage, V14 refers to 14 leaf stage, R1 refers to silking and
R2 refers to blister stage of the corn. (Details are highlighted in 2.1.2. Spectrum
acquisition (Line 88-89) and
https://pan.baidu.com/s/1YIR5GG7nip7brCekIG0VGw of the revised manuscript)
Point 3、Table 2.- What does C.V. stand for? It must be explained somewhere.
Response: C.V. refers to the coefficient of variation. (Details are in Table 1 (Line 108-
109) of the revised manuscript)
Point 4、Lines 131-132: the authors say that they used data from 72 experimental plots.
They have not explained how many of these plots correspond to each corn species, and
how many of these plots correspond to the V6, V14, R1 and R2 seasons. They also have
to explain the extension of each of the plots, and how they were chosen. I think this part
of the experiment has to be explained in more detail.
Response: I am sorry that we used 18 study plots. 1# - 9# plots were planted ‘Nongda
108’ (Mid-drape type) and 10# - 18# plots were planted ‘Jinghua 8’ (compact type).
The total study area was 924 m 2 and each area was 7×7 m 2 . Corn samples were
extracted at four growth stages of V6, V14, R1 and R2 in 18 plots, respectively. (Details
are highlighted in Figures 1 (Line 84-86) and 2.1.1. Study area (Line 78-82) of the
revised manuscript)
Point 5、Regarding Spectral position features: how do the authors calculate the base
lines to calculate the absorption depth and the reflection depth?
Response: This is a good question. The three absorptions (560–760, 920–1080, and
1120–1280 nm) and six reflections (500–670, 780–970, 980–1200, 1200–1350, 1480–
1720, and 2000–2300 nm) that were used to study the characteristic absorption and
reflection positions.
1) The absorption depth is the difference between unity and continuum-removal
reflectance ? ′ ?
(?
min ). ? ′ ?
(?
min ) is defined as the ratio of ? ?
(?
min ) that is the
reflectance at corresponding wavelength λ in the absorption region to the outer
continuum line ? ??
(?
min ) in the corresponding band. The calculate is:
A_Depth ? = 1 − ? ′ ?
(?
min ) = 1 − ? ?
(?
min )/? ??
(?
min ).
2) The reflection depth is the difference between unity and the continuum-removed
reflectance ? ′ ?
(?
max
).
The continuum-removed reflectance ? ′ ?
(?
max ) is the ratio
of the inner continuous line ? ??
(?
max ) in the reflection position and the maximum
reflectance value ? ?
(?
max ) at the corresponding band. The calculate is:
R_Depth ? = 1 − ? ′ ?
(?
max ) = 1 − ? ??
(?
max )/? ?
(?
max
)
(Details are highlighted in References 16,17,18 (Line 442-448) and 2.2.2. Spectral
position features (Line 133-138,148-153) of the revised manuscript)
Point 6、Reganding 10) Spectral position features: in fig. 2, I guess the dashed lines
are the base lines used to calculate the absorption depth and the reflection depth. Please,
in the figure caption the meaning of the dashed lines should be given.
Response: The red dotted line was the inner continuous line ? ??
(?
max ) in the
reflection position, the blue dotted line was outer continuum line ? ??
(?
min ) in the
absorption region. (Details are in Figure 2 (Line 130-132) of the revised manuscript)
Point 7、Figure 2. The caption of this figure includes some text in Chinese that should
be translated into English or deleted.
Response: I have been translated the caption of this figure into English. (Details are in
Figure 2 (Line 127-128) of the revised manuscript)
Point 8、Line 217-218: The authors state: “The number with minimum RMSE was the
number of variables”: I do not understand the meaning of this sentence. What does this
have to do with how SPA works?
Response: This is a good question. In order to solve the collinearity problems, a
minimally redundant subset of wavebands is selected in SPA and it is also a forward
selection method. SPA starts with one wavelength and selects a new one at each iteration
by using projection operators in a vector space until reaching the predefined number of
wavelengths. The root means square error (RMSE) was used as the evaluation index.
The number was the number of variables while the RMSE was minimum. This is a
condition of the SPA that select the sensitive variables of hyperspectral data. (Details
are highlighted in 2.3.1. Successive projections algorithm (Line 193-197) of the
revised manuscript)
Point 10、Line 284-285: the authors state that “The SPA algorithm screened out six
sensitive reflection spectra with the least linearity of leaf nitrogen content, and six
sensitive spectra were modeled by PLS and RF regression”. Do they really mean six
sensitive reflection spectra? I do not understand how SPA is used, I thought it was being
used to screen out spectral characteristics, not spectra. Furthermore, if only six spectra
were screened out to be used with PLS and RF, why is it that they used 48 spectra for
calibration and 24 for validation, as shown in Table 5. This is not clear at all.
Response: I am sorry for the trouble that I bring to you. When the RMSE is minimum,
the number is the number of variables. This is a condition of the SPA to select sensitive
variables of hyperspectral data. The SPA can select six sensitive spectra (including four
reflectance spectra and two first derivative spectra) and between four reflectance
spectra and LNC established a model of the PLS and RF regression. Similarly, between
two first derivative spectra and LNC established a model of the PLS and RF regression.
Because the study has 18 plots and four growth stage, the total data is 72. We use 48 to
calibration and 24 for validation. The results show in Table 4. (Details are highlighted
in 3.1.1. Sensitive spectral feature data set (Line 264-268) and Table 4 (Line 283-284)
of the revised manuscript)
Point 11、Another point is that if SPA is used to screen out spectral characteristics, the
author must give the six spectral characteristics that were used in PLS and RF.
Response: The SPA can select six sensitive spectra (including four reflectance spectra:
412, 724,1084,1343 nm and two first derivative spectra: 658, 937 nm). Between four
reflectance spectra and LNC established a model of the PLS and RF regression.
Similarly, between two first derivative spectra and LNC established a model of the PLS
and RF regression. (Details are highlighted in 3.1.1. Sensitive spectral feature data set
(Line 264-268) and Table 4 (Line 283-284) of the revised manuscript)
Point 13、Line 363-364: The authors state that: “Applying the SPA algorithm was
selected three optimal sensitive variables of hyperspectral data”. But on Line 340 they
state that 4 spectral characteristics were selected using SPA: “The analysis showed that
the spectral reflection bands at 724 and 1343 nm, FD 340 band at 658 nm, and NDVIg-
b” This makes 4 spectral characteristics. My question is: how many spectral
characteristics were chosen: three or four?
Response: I select three optimal spectral characteristics: sensitive spectral feature data
set, position feature data set and vegetation indices data set. But the composite spectral
features are selected from three optimal spectral characteristics. That means we use the
SPA twice in the composite spectral features. (Details are highlighted in 3.1.1. Sensitive
spectral feature data set (Line 264-267), 3.1.2. Position feature data set (Line 315-
316), 3.1.3. Vegetation indices data set (Line 290-291) and 3.2. Composite spectral
features (Line 306-307) of the revised manuscript)
Point 13、As for the English language, there are a lot of sentences that are not correctly
written, and verbal tenses are not correctly used. Some examples:
Line 189: edit
Line 190: “summarized”
Line 212: “had been became”
Line 213: “SPA was”
Line 220: “was”, “combined”
Line 264: “revealed”
Lien 265: “showed”
Line 286: “were”
Line 336: “improved”
Line 358-359: “were a poor correlation”
Linde 363-364: Applying the SPA algorithm was selected three optimal sensitive
variables of hyperspectral data”
Response: Thank you for your suggestion. I have revised the sentences that are not
correctly written.
Line 189: edit, I am sorry I am not found it.
Line 164: “summarized” to “showed”
Line 186: “had been became” to “has been”
Line 187: “SPA was” to “and is”
Line 199: “was”, “combined” to “is” “included”
Line 243: “revealed” to “showed”
Line 244: “showed” to “suggested”
Line 293: “were” delete
Line 317: “improved” to “improve”
Line 366: “were a poor correlation” were delete
Line 371: “Applying the SPA algorithm was selected three optimal sensitive variables
of hyperspectral data” were delete

Round 2
Reviewer 1 Report
The manuscript presents a methodology for estimating leaf nitrogen content (LNC) using hyperspectral data obtained through a handheld spectroradiometer.
Although some improvements were introduced in the revised version of the manuscript, an extensive revision of English language and style remains necessary throughout the manuscript.
The Introduction and Discussion sections need to be revised. Although the Authors made an effort to include additional information, a clarification of the ideas and contents is necessary.
Specific comments are presented below:
Introduction
Lines 8 – 45 - Although there are some improvements in the Introduction section, there is still a lack of focus in the presentation of the 3 categories defined by the Authors for the hyperspectral data. The Authors describe a set of applications relative to the 3 categories of data, some related with LNC, others related with chlorophyll content, others related with LAI... the information about these applications seems random and without a logic. Although the various types of applications should be mentioned, it's important focusing on:
- Have these 3 categories of data been used (individually) for estimating LNC?
- If so, those applications should be emphasized. If not, that should be said and presented as a gap.
Line 15 – Replace LCC by LNC
Line 16 - A brief description of position of reflected features should be provided.
Material and Methods
Lines 193 – 194 –The SPA is based on a forward selection method or it was used SPA plus a forward selection method? Please clarify and revise the sentence accordingly.
Line 197 – I suggest replacing the sentence by: “The final number of variables selected by the SPA is defined based on the lower RMSE value obtained”.
Results
Replace lines 263-267 by: "The SPA screened out six sensitive spectra (including four reflectance spectra:412, 724,1084,1343 nm and two first derivative spectra: 658, 937 nm) with the least linearity of leaf nitrogen content. A model was established between the four reflectance spectra and the LNC based on PLS and RF regression. Similarly, a model was established between the two first derivative spectra and LNC based on PLS and RF regression."
Lines 263 – 267 - The Authors' reply provided in the cover letter to Point 4 (relative to the reason why the reflection spectral bands and the first derivative bands selected were used separately in PLS and RF models) should be included here in the manuscript text.
Lines 301 - "...weaker relevance" instead of "...weaker relevant"
Lines 302 - Replace “Because the circles of this VIs were small and the color were light. “ by “Such results are expressed in Figure 6, where these VIs are represented by smaller circles and lighter color when compared to the other VIs.”
Discussion
This section needs to be revised. Although the Authors made an effort to include additional information, a clarification of the ideas and English language is necessary.
Lines 336-340 – The sentence needs revision because is not clear what the Authors mean to say.
Line 340 - 724 instead of 72
Line 346 - The sentence "Because serious multi-collinearity problem arose in sensitive bands, positions and VIs." needs revision. I suggest removing "Because"
Lines 346 – 347 - What are the three optimal sensitive variables? Please specify.
Lines 348 – 349 - Replace "but between the position features and LNC are bad than the other" by "but the correlation between the position features and LNC is low"
Lines 357 – 358 – The sentence “The R2 of the PLS was increased 0.05 in validation set than Chen et al. [66] and the R2 of the PLS model was not much different” needs revision.
What the Authors mean to say? The value of R2 obtained in the present study was larger than the value obtained by Chen et al. [66]? And also, the “R2 of the PLS was not much different” from what?
Lines 358 – 360 – Sentence needs revision. Why the Authors start the sentence with “Because”? Instead I suggest writing: “The composite spectral features integrated characteristics of three variable sets and the results of PLS model were more stable, when comparing the results for calibration and validation datasets, than any other three variable datasets used independently.”
Line 360 – “reflectance bands” instead of “reflect bands”
Conclusion
Lines 380 and 381 – The Authors say “LNC had better correlation with these three types of spectral variables.” If you say that something is better you have to indicate what are you comparing to.
Also, you should clearly indicate that the results are better, when considering the results both from calibration and validation.
Author Response
Manuscript ID: sensors-519557
Title: Hyperspectral-based Estimation of Nitrogen Content in Corn Leaves Using
Optimal Selection of Multiple Spectral Variables
Author: Lingling Fan, Jinling Zhao*, Xingang Xu, Dong Liang, Guijun Yang, Haikuan
Feng, Hao Yang, Yulong Wang, Guo Chen, Penfei Wei
Original Research
Scientific Reports
Cover letter
Dear reviewer,
Thank you very much for your reply and reviewers’ constructive suggestions for
improving our manuscript (sensors-519557). We revised the manuscript point by point
very carefully according to the reviewers’ suggestions. Attached please find the revised
manuscript and the following responses to reviewers’ comments. In the manuscript, all
revised parts are highlighted in red for convenience of being reviewed by you and
reviewers.
With best regards,
Lingling Fan and co-authors
****************************************************
Correspondence information: Jinling Zhao
National Engineering Research Center for Agro-Ecological Big Data Analysis &
Application, Anhui University, Hefei 230601, China.
E-mail: [email protected]
Telephone: +(0551) 6295 0280
****************************************************
Response to Reviewer 1 Comments
Point 1:Although some improvements were introduced in the revised version of the
manuscript, an extensive revision of English language and style remains necessary
throughout the manuscript.
Response: We have revised the language mistakes carefully the attached PDF file.
Thank for your good suggestion very much. (The revised details are corresponding to
the PDF file from the reviewer)
Point 2: Have these 3 categories of data been used (individually) for estimating LNC?
If so, those applications should be emphasized. If not, that should be said and presented
as a gap.
Response: This is a good question. Not all references are used to estimate LNC
separately and it focuses on one aspect. Meanwhile, I have emphasized those
applications. (Details are highlighted in References 3,6 (Line 454-455, Line 463-464)
of the revised manuscript)
Point 3: Line 15 – Replace LCC by LNC.
Response: Thank you for your suggestion. I have revised the sentence. (Details are
highlighted in 1. Introduction (Line 54) of the revised manuscript)
Point 4: Line 16 - A brief description of position of reflected features should be
provided.
Response: Thank you for your suggestion. I have revised the content of this part.
(Details are highlighted in 1. Introduction (Line 55-62) of the revised manuscript)
Point 5: Lines 193 – 194 –The SPA is based on a forward selection method or it was
used SPA plus a forward selection method? Please clarify and revise the sentence
accordingly.
Response: The SPA belongs to the class of forward selection methods. (Details are
highlighted in 2.3.1. Successive projections algorithm (Line 227) of the revised
manuscript)
Point 6: Line 197 – I suggest replacing the sentence by: “The final number of variables
selected by the SPA is defined based on the lower RMSE value obtained”.
Response: Thank you for your suggestion. I have replaced the sentence with "The final
number of variables selected by the SPA is defined based on the lower RMSE value
obtained". (Details are highlighted in 2.3.1. Successive projections algorithm (Line
230-231) of the revised manuscript)
Point 7: Replace lines 263-267 by: "The SPA screened out six sensitive spectra
(including four reflectance spectra:412, 724,1084,1343 nm and two first derivative
spectra: 658, 937 nm) with the least linearity of leaf nitrogen content. A model was
established between the four reflectance spectra and the LNC based on PLS and RF
regression. Similarly, a model was established between the two first derivative spectra
and LNC based on PLS and RF regression."
Response: Thank you for your suggestion. I have replaced the sentence with "The SPA
screened out six sensitive spectra (including four reflectance spectra:412,
724,1084,1343 nm and two first derivative spectra: 658, 937 nm) with the least linearity
of leaf nitrogen content. A model was established between the four reflectance spectra
and the LNC based on PLS and RF regression. Similarly, a model was established
between the two first derivative spectra and LNC based on PLS and RF regression."
(Details are highlighted in 3.1.1. Sensitive reflectance feature data set (Line 302-306)
of the revised manuscript)
Point 8: Lines 263 – 267 - The Authors' reply provided in the cover letter to Point 4
(relative to the reason why the reflection spectral bands and the first derivative bands
selected were used separately in PLS and RF models) should be included here in the
manuscript text.
Response: Thank you for your suggestion. I've put the point 4 of the cover letter into
my manuscript. (Details are highlighted in 3.1.1. Sensitive reflectance feature data set
(Line 297-301) of the revised manuscript)
Point 9: Lines 301 - "...weaker relevance" instead of "...weaker relevant".
Response: Thank you for your suggestion. I have replaced the sentence. (Details are
highlighted in 3.1.3. Vegetation indices data set (Line 339) of the revised manuscript)
Point 10: Lines 302 - Replace “Because the circles of this VIs were small and the color
were light. " by “Such results are expressed in Figure 6, where these VIs are represented
by smaller circles and lighter color when compared to the other VIs. "
Response: Thank you for your suggestion. I have replaced the sentence. (Details are
highlighted in 3.1.3. Vegetation indices data set (Line 340-341) of the revised
manuscript)
Point 11: Lines 336-340 – The sentence needs revision because is not clear what the
Authors mean to say.
Response: I'm sorry for the trouble I have caused you. In order to reduce the influence
of water vapor and other factors of hyperspectral data, we chose 400-1353 nm, 1437-
1799 nm and 1992-2354 nm to study the spectra. Firstly, the range of selected bands is
introduced, and then the sensitive variables constructed are described in this range. The
following is cited. (Details are highlighted in 4. Discussion (Line 376-377) of the
revised manuscript)
Point 12: Line 340 - 724 instead of 72
Response: Thank you for your suggestion. I have replaced the sentence. (Details are
highlighted in 4. Discussion (Line 378) of the revised manuscript)
Point 13: Line 346 - The sentence "Because serious multi-collinearity problem arose
in sensitive bands, positions and VIs." needs revision. I suggest removing "Because".
Response: Thank you for your suggestion. I have removed "Because". (Details are
highlighted in 4. Discussion (Line 385) of the revised manuscript)
Point 14: Lines 346 – 347 - What are the three optimal sensitive variables? Please
specify.
Response: This is a good suggestion. The three optimal sensitive variables are the
optimal sensitive band features, the optimal sensitive position features and the optimal
sensitive VIs. (Details are highlighted in 4. Discussion (Line 385-386) of the revised
manuscript)
Point 15: Lines 348 – 349 - Replace "but between the position features and LNC are
bad than the other" by "but the correlation between the position features and LNC is
low".
Response: Thank you for your suggestion. I have replaced the sentence. (Details are
highlighted in 4. Discussion (Line 387-388) of the revised manuscript)
Point 16: Lines 357 – 358 – The sentence "The R2 of the PLS was increased 0.05 in
validation set than Chen et al. [66] and the R2 of the PLS model was not much different"
needs revision.
What the Authors mean to say? The value of R2 obtained in the present study was larger
than the value obtained by Chen et al. [66]? And also, the “R2 of the PLS was not much
different” from what?
Response: Chen et al. suggested that the R 2 values were 0.72 for corn. And ours results
were increased 0.05 than theirs in the PLS model. (Details are highlighted in 4.
Discussion (Line 397-399) of the revised manuscript)
Point 17: Lines 358 – 360 – Sentence needs revision. Why the Authors start the
sentence with “Because”? Instead I suggest writing: “The composite spectral features
integrated characteristics of three variable sets and the results of PLS model were more
stable, when comparing the results for calibration and validation datasets, than any
other three variable datasets used independently.”
Response: Thank you for your suggestion. I have replaced the sentence. (Details are
highlighted in 4. Discussion (Line 399-402) of the revised manuscript)
Point 18: Line 360 – “reflectance bands” instead of “reflect bands”.
Response: Thank you for your suggestion. I have replaced the sentence. (Details are
highlighted in 4. Discussion (Line 402) of the revised manuscript)
Point 19: Lines 380 and 381 – The Authors say “LNC had better correlation with these
three types of spectral variables.” If you say that something is better you have to
indicate what are you comparing to.
Also, you should clearly indicate that the results are better, when considering the results
both from calibration and validation.
Response: Thank you for your suggestion. I have revised the sentence that "LNC had
better correlation with the optimal sensitive bands and VIs (Figure 4,5), but the optimal
positions have a bad correlation with LNC (Figure 6) ". (Details are highlighted in 5.
Conclusion (Line 422-424) of the revised manuscript)

Reviewer 2 Report
I think the authors have improved the paper according to the referee's comments.
However, there are still 3 points that should be addressed before it is accepted for publication:
1) Lines 263-264. I am afraid that when the authors say "reflectance spectra" in fact they mean "reflectance spectral wavelength " or "reflectance feature". The same when they refer to "derivative spectra" they mena "derivative spectral wavelenghts" or "derivative spectra features". Please this confusion must be corrected.
2) Line 197. The explanation of SPA is rather confusing. When they explain how it works, on line 197 they should state clearly that "the number of features finally selected is that that minimizes RMSE".
3) Some verbs are not used in the correct tense. There are plenty of examples throughout the manuscripts. When theh authors refer to the results shown in tables in figures, the present tenxe is preferred, while they insist on using the past simple. I recommend to revise the whole manuscript.
Author Response
Manuscript ID: sensors-519557
Title: Hyperspectral-based Estimation of Nitrogen Content in Corn Leaves Using
Optimal Selection of Multiple Spectral Variables
Author: Lingling Fan, Jinling Zhao*, Xingang Xu, Dong Liang, Guijun Yang, Haikuan
Feng, Hao Yang, Yulong Wang, Guo Chen, Penfei Wei
Original Research
Scientific Reports
Cover letter
Dear reviewer,
Thank you very much for your reply and reviewers’ constructive suggestions for
improving our manuscript (sensors-519557). We revised the manuscript point by point
very carefully according to the reviewers’ suggestions. Attached please find the revised
manuscript and the following responses to reviewers’ comments. In the manuscript, all
revised parts are highlighted in red for convenience of being reviewed by you and
reviewers.
With best regards,
Lingling Fan and co-authors
****************************************************
Correspondence information: Jinling Zhao
National Engineering Research Center for Agro-Ecological Big Data Analysis &
Application, Anhui University, Hefei 230601, China.
E-mail: [email protected]
Telephone: +(0551) 6295 0280
****************************************************
Response to Reviewer 2 Comments
Point 1: Lines 263-264. I am afraid that when the authors say "reflectance spectra" in
fact they mean "reflectance spectral wavelength " or "reflectance feature". The same
when they refer to "derivative spectra" they mean "derivative spectral wavelengths" or
"derivative spectra features". Please this confusion must be corrected.
Response: Thank you for your suggestion. I have revised the sentences. (Details are
highlighted in 3.1.3. Vegetation indices data set (Line 302-303) of the revised
manuscript)
Point 2: Line 197. The explanation of SPA is rather confusing. When they explain how
it works, on line 197 they should state clearly that "the number of features finally
selected is that that minimizes RMSE".
Response: Thank you for your suggestion and your advice is good. I have revised the
sentence with " The final number of variables selected by the SPA is defined based on
the lower RMSE value obtained. " according to the first reviewer. (Details are
highlighted in 2.3.1. Successive projections algorithm (Line 230-231) of the revised
manuscript)
Point 2: Some verbs are not used in the correct tense. There are plenty of examples
throughout the manuscripts. When the authors refer to the results shown in tables
in figures, the present tense is preferred, while they insist on using the past simple. I
recommend to revise the whole manuscript.
Response: We have revised the language tense carefully the attached PDF file. Thank
for your good suggestion very much. (The revised details are corresponding to the PDF
file from the reviewer)
